# Child and Parent Perceived Determinants of Children’s Inadequate Sleep Health. A Concept Mapping Study

**DOI:** 10.3390/ijerph17051583

**Published:** 2020-02-29

**Authors:** Laura S. Belmon, Vincent Busch, Maartje M. van Stralen, Dominique P.M. Stijnman, Lisan M. Hidding, Irene A. Harmsen, Mai J.M. Chinapaw

**Affiliations:** 1Amsterdam UMC, Vrije Universiteit Amsterdam, Department of Public and Occupational Health, Amsterdam Public Health Research Institute, 1081 BT Amsterdam, The Netherlands; 2Sarphati Amsterdam, Public Health Service (GGD), 1018 WT Amsterdam, The Netherlands; 3Department of Epidemiology and Health Promotion, Section Youth, Municipal Health Service Amsterdam, 1018 WT Amsterdam, The Netherlands; 4Department of Health Sciences, Faculty of Science and Amsterdam Public Health Research Institute, Vrije Universiteit Amsterdam, 1081 HV Amsterdam, The Netherlands

**Keywords:** sleep, childhood, children, determinants, factors, concept mapping

## Abstract

Many children do not meet the recommendations for healthy sleep, which is concerning given the potential negative effects on children’s health. To promote healthy sleep, it is crucial to understand its determinants. This concept mapping study therefore explores perspectives of children and parents on potential determinants of children’s inadequate sleep. The focus lies on 9–12 year old children (*n* = 45), and their parents (*n* = 33), from low socioeconomic neighbourhoods, as these children run a higher risk of living in a sleep-disturbing environment (e.g., worries, noise). All participants generated potential reasons (i.e., ideas) for children’s inadequate sleep. Next, participants sorted all ideas by relatedness and rated their importance. Subsequently, multidimensional scaling and hierarchical cluster analyses were performed to create clusters of ideas for children and parents separately. Children and parents both identified psychological (i.e., fear, affective state, stressful situation), social environmental (i.e., sleep schedule, family sleep habits), behavioural (i.e., screen behaviour, physical activity, diet), physical environmental (i.e., sleep environment such as temperature, noise, light), and physiological (i.e., physical well-being) determinants. These insights may be valuable for the development of future healthy sleep interventions.

## 1. Introduction

For children, healthy sleep is typically defined as a regular sleep rhythm consisting of approximately 9–11 hours per night [1] of good quality sleep (i.e., the combination of high sleep efficiency and a good subjective assessment of their own sleep) [2]. This nightly dose of healthy sleep is important for children’s cognitive performance [3,4], academic performance [3,5,6,7], and their physical and mental health [5,8,9,10]. Unfortunately, children’s average sleep duration has declined significantly in recent decades [11], while problems such as daytime sleepiness and longer sleep onset latency have become more prevalent [12,13]. This is especially apparent among children from lower socioeconomic (low SEP) neighbourhoods [14]. This may be ascribed to various environmental factors, such as noise nuisance caused by cramped housing or more parental stress caused by financial problems.

Knowledge of the most relevant determinants is essential for the development of effective interventions [15]. A recent review of prospective studies [16] found evidence that spending more time on screens (e.g., TV, computer, games), having a difficult temperament, and past poor or inadequate sleep health (i.e., sleep quality or quantity) were longitudinally associated with shorter sleep duration. Another review summarized empirical evidence related to common paediatric sleep recommendations and identified having an inappropriate bedtime, not having a relaxing bedtime routine, having an irregular sleep schedule, a negative emotional environment (e.g., family stress, family conflict), and poorer emotional well-being (e.g., higher levels of internalizing symptoms) as potential determinants of children’s inadequate sleep [17]. However, the perspectives of children and their parents are lacking in the current literature. 

The perspectives of children and their parents could bring about new and important insights into potential determinants of inadequate sleep, which can subsequently inform intervention development. Consequently, the aim of this study is to explore the perspectives of children and parents living in low-SEP neighbourhoods on potential determinants of children’s inadequate sleep health. 

## 2. Materials and Methods 

A participatory mixed-methods concept mapping study was conducted to assess children’s and parents’ perspectives on potential determinants of children’s inadequate sleep health [18]. For the qualitative part of this approach, participants generated ideas about potential determinants during group brainstorm sessions, and subsequently rated these ideas according to importance. Researchers were not allowed to add or prompt additional ideas. Concept mapping is a 6-step process (see Figure 1). The first five steps are illustrated below. Step six includes the identification of perceived determinants that might be included in future healthy sleep interventions (see Discussion). Concept mapping has been used in previous studies into children’s perspectives on behavioural determinants [19,20]. Additional information about the concept mapping approach can be found elsewhere [18].

### 2.1. Preparation (Step 1)

The preparation phase included providing focus for the conceptualization, followed by identifying and recruiting participants. 

#### 2.1.1. Providing Focus for the Conceptualization

The first step was the creation of a ‘*focus statement*’ and ‘*rating statement*’: a main question or statement that gives a specific instruction for the session [18]. The purpose of the focus statement is to elicit ideas about the topic of interest, whereas the rating statement provides comparative ratings of importance for the generated ideas (see Figure 1). 

The comprehensiveness of the statements and the feasibility of the sorting and rating task were tested in a pilot study. For this, two pilot sessions were conducted with parents (9 and 6 parents, respectively) and two with children (12 and 26 children, respectively), and changes were made to clarify the focus- and rating statements. 

#### 2.1.2. Participants and Recruitment 

Children and parents were recruited through schools and thereby grouped based on school level and availability. The health advisors of the Public Health Service of Amsterdam brought the researchers in contact with primary schools in socioeconomically disadvantaged neighbourhoods, based on the postal code of the neighbourhood [21]. Four out of 23 invited primary schools participated. When schools were willing to participate, they were asked to distribute information letters to children aged 9–12 years old and parents with at least one child in the age range 4–12 years. The age range of the children (i.e., 9–12 years) was chosen because the tasks within the study were considered conceptually too difficult for children younger than 9 years old [22]. Participating children and at least one parent/caregiver provided informed consent. Schools were offered a lecture about healthy sleep as an incentive for participation, while children received a small present and parents a gift card with a value of 10 euros. The VU University Medical Ethical Committee approved the study protocol and concluded that it does not fall within the scope of the Medical Research Involving Human Subjects Act (study protocol 2017.013).

### 2.2. Generation of Ideas (Step 2)

At each primary school, two concept mapping sessions (1 to 1.5 hours) were organized per subgroup of children or parents, with approximately one week between sessions. The sessions were facilitated by one researcher (L.S.B.) and assisted by a second researcher (M.M.v.S, V.B., I.A.H., E.M.R., E.E.V., A.W., R.P. or L.B.). Each first session consisted of generating ideas in a brainstorm session. In each session, participants were first given a ‘warm-up question’ to stimulate understanding of the concept: “*What is inadequate sleep for you / for children?*”. Following this, they were encouraged to brainstorm individually about the *focus statement* (see Figure 1) and write down as many ideas as possible. Subsequently, everyone shared their ideas, one by one, with the rest of the group until none were left unmentioned, resulting in a complete list of original ideas per subgroup. During the first session, participants also completed a short questionnaire asking them about age, gender, education level (parents), and perceived cultural group (parents) or country of birth (children). Parents’ education categories were defined as 1) low, i.e., highest education level is primary or secondary school education or no education at all; 2) medium, i.e., highest education is secondary vocational education; and 3) high, i.e., highest education is higher professional education or scientific education. After each brainstorm session, ideas that were conceptually the same were merged, resulting in a set of unique ideas that were printed on cards for the second session. One researcher (L.S.B.) suggested the adaptations, which were checked by a second researcher (I.A.H.). In case of disagreement, a third researcher (V.B. or M.M.v.S.) was consulted. 

### 2.3. Structuring Ideas (Step 3) 

The second session consisted of structuring ideas. Firstly, participants individually sorted (i.e., clustered) all ideas by relatedness. The rules for this task were; 1) create a minimum of three and a maximum of 10 piles; 2) all cards (i.e., ideas) need to be placed on a pile; 3) a pile cannot consist of a single card (i.e., one idea); and 4) there cannot be a miscellaneous pile [18,19]. The last step of the sorting task was to name their piles of ideas. The name of each pile had to represent the relatedness between ideas. Secondly, they individually rated all ideas according to the rating statement: “*Think about your sleep, how much does this affect your sleep? / Think about the sleep of a child in the age of 4–12 years, how much does this affect their sleep?”* on a Likert scale: (1) ‘does not affect at all’ represented by a rested emoticon; (2) hardly affects; (3) affects a little; (4) affects a lot; (5) ‘affects a whole lot’ represented by a very tired emoticon [18]. 

### 2.4. Analyses (Step 4)

The software programme Ariadne [23] was used for the data analyses; multidimensional scaling, and hierarchical cluster analyses per subgroup [18]. We created a ‘two-dimensional point map’ on which each point represents an individual idea, with a specific ‘distance’ to the other ideas. Points that lie close to each other on the map represent ideas that were grouped together more often by participants. The table of similarities (or similarity matrix) structures the information of each participant about their perception of the relationship between ideas (i.e., the sorting task). 

### 2.5. Interpretation (Step 5)

Two researchers (L.S.B. and D.P.M.S.) independently determined the optimal number of clusters for each subgroup using the divisive method and ‘hierarchical cluster tree’. For this method, all ideas start off in a single cluster. Based on the individual clustering of participants, the programme subsequently suggests how the ideas can be optimally arranged into clusters when choosing two, three, four, or more clusters. The underlying ideas in each cluster were interpreted incrementally and critically reviewed until each idea was in a cluster with other ideas that reflected a similar concept [18]. Potential conflicts were resolved by a third researcher (V.B.). Some ideas were moved to another cluster or a new cluster was created when this made more sense conceptually. These decisions can be found in Appendix A (Figure A1, Figure A2, Figure A3, Figure A4, Figure A5, Figure A6, Figure A7, Figure A8, Figure A9 and Figure A10) and B (Table A1, Table A2, Table A3, Table A4, Table A5, Table A6, Table A7, Table A8, Table A9 and Table A10). This process continued until consensus was reached on the number and meaning of the clusters per subgroup. After this, the clusters were named based on participants’ input. One concept map was created per subgroup. For the final concept maps see Appendix A. 

Clusters that represented multiple topics (i.e., perceived determinants) were split up. The original ideas were merged into main ideas for children and parents separately (see Appendix B). The average importance rating of each perceived determinant was calculated based on the average importance ratings of the underlying main ideas, and the average importance rating of each main idea was based on the average importance ratings of the underlying original ideas. An average ‘overall importance rating’ for all perceived determinants and main ideas was calculated by combining the mean rating of all participants across all groups, for children and parents separately. This created a clear overview of the importance of the perceived determinants and main ideas across all subgroups. An average rating of ≥3.00 was considered as important. Ratings between 2.95 and 2.99 were rounded to 2.9.

## 3. Results

### 3.1. Participants

The education level of the participating parents was mainly medium (42.4%) and high (42.4%). Six groups of children (*N* = 5 to 9) and four groups of parents (*N* = 7 to 10) were formed. Table 1 presents the sample characteristics.

### 3.2. Concept Maps

Children generated 30 to 58 ideas per group and parents 32 to 58 ideas. Some ideas generated in the first session were perceived as unclear or difficult in the subsequent session. In this case, the idea was excluded from sorting and rating (e.g., children in group 1 eventually discarded the idea ‘dancing in my bedroom at night’ and group 2 did so with ‘sleepwalking’). Two of the youngest children in group 1 experienced the second session as difficult and their clustering and ratings were therefore excluded from the analyses. The final number of clusters defined by the researchers ranged from four to six, for both children and parents. As the majority of these clusters represented multiple perceived determinants, we separated them to provide a clear overview of the different perceived determinants and their mean ratings. Table 2 presents children’s perceived determinants and the underlying ideas, categorized in five determinant domains: psychological, physiological, physical environmental, social environmental, and behavioural. Table 3 presents parents’ perceived determinants. In both tables, the determinants and underlying ideas are sorted from high to low importance.

### 3.3. Perceived Determinants of Children’s Inadequate Sleep

None of the child-perceived determinants were rated as important (≥3.00) by the children with an average score ranging from 1.9 to 2.9. However, two of the underlying main ideas were rated as important (≥3.00) and were mentioned by all groups of children: nightmares and illness. Other underlying main ideas that were mentioned by all groups of children but had a lower overall rating for importance were: noise from inside the house, distractions in the bedroom, excitement, negative affective state, and a recent stressful event. In addition, the underlying main ideas that did have a high importance score but were only mentioned in some groups of children were: a recent scary event, scary thoughts, not the right temperature, not being able to lie down comfortably, having too many thoughts, an upcoming stressful event, not being tired and going to bed too early. 

Almost all parent-perceived determinants and their underlying main ideas were rated as important (≥3.00) ranging from 2.9 to 3.9, and 2.1 to 4.4, respectively. The underlying main ideas that were mentioned by all groups of parents were: illness, parental relationship problems, being bullied, an upcoming stressful event, a recent stressful event, no consistent sleep schedule, watching something scary, screen use before bedtime, deviating from bedtime routine. The underlying main idea with the highest importance rating was feeling unsafe (mean rating of 4.4).

Most of the determinants mentioned by parents were also mentioned by children. However, the same ideas were often clustered differently by parents and children: 1) ‘physical well-being’ (parents) versus ‘discomfort’ and ‘diet’ (children); 2) ‘affective state’ and ‘stressful situation’ (parents) versus ‘affective state’ (children); 3) ‘energy’ and ‘physical activity’ (parents) versus energy (children); 4) ’sleep schedule’; 5) ‘sleep environment’; and 6) ‘activating activities’ (parents) versus ‘screen behaviour’ (children). Potential determinants that were mentioned by parents but not by children were ‘family sleep habits’ and ‘social environment’.

## 4. Discussion

The aim of this study was to explore potential determinants of children’s inadequate sleep health, from the perspective of school-aged children and parents living in low-SEP neighbourhoods. These perspectives brought about important insights, which can inform future intervention development to promote healthy sleep. Both children and parents identified various potential determinants of children’s inadequate sleep health which were categorized into psychological (i.e., fear, affective state), social environmental (e.g., sleep schedule, stressful situation), behavioural (e.g., screen behaviour, physical activity, diet), physical environmental (i.e., sleep environment), and physiological (i.e., discomfort, physical well-being) determinants. 

Fear was rated by children as most important. They described fear due to a recent scary event, reading or watching something scary, hearing scary sounds, or having a nightmare. Parents also rated affective state, including being afraid, as important. A recent review (2019) [16] found inconclusive evidence for a relationship between anxiety symptoms and sleep duration, and no evidence for a relationship with sleep quality. However, the results in this review are based on only two longitudinal studies. Additionally, Bagley et al. (2015) found that pre-sleep worries mediated the relationship between family income and children’s sleep health based on cross-sectional data [24]. Potentially, those who experienced anxiety engaged in unhelpful pre-bedtime behaviours [25]. Both children and parents in the current study rated the perceived psychological determinants, fear and affective state, as important, and these determinants included many underlying ideas. Consequently, dealing with fear or other negative affective feelings before falling asleep, concurrent with good sleep hygiene practices, may be a promising focus when promoting healthy sleep. This may be included in healthy sleep interventions by teaching children relaxation techniques, such as meditation, breathing exercises, imagination journeys [26] in both school, after-school and community settings. Additionally, teaching parents (i.e., online or through community programmes) how to implement such techniques in their child’s bedtime routine could be an important avenue for promoting healthy sleep.

Both children and parents identified a stressful situation (e.g., being bullied, parental stress, and parental relationship problems) as a potential determinant of inadequate sleep. This aligns with the results of a review that concluded that children who live in a supportive family environment and have a healthy relationship with their parents generally sleep longer and more efficiently [27]. A supportive and healthy family environment is characterized by parents’ involvement in their child’s life, and a good relationship between the child and its caregivers [27]. This also means parents should be aware of whether their child is going through a difficult time and support their child when needed (e.g., when the child has had a negative experience such as being bullied). Furthermore, one longitudinal study found that parent-child physical conflict (i.e., verbal aggression such as screaming, and physical aggression such as beating) was a determinant of children’s insufficient sleep duration [28]. Our findings support existing evidence that creating a positive and supportive family environment is important for healthy sleep. It may therefore be valuable to encourage parents to monitor their child’s needs, desires and stressors. Such interventions may include letting parents re-evaluate their behaviour, giving scenario-based risk information, or raising consciousness to increase parents’ awareness [15]. Increasing awareness must be quickly followed by increasing parent’s problem-solving ability and self-efficacy (i.e., confidence in their ability) by using methods such as goal setting, self-monitoring of behaviour, setting graded tasks, and planning coping responses [15]. This may reach parents via face-to-face or web-based sessions.

An inadequate sleep schedule, including an inappropriate bedtime (i.e., too early or late), an inconsistent sleep schedule (i.e., varying bedtimes throughout the week and weekend) and daytime napping, was identified by both children and parents as determinant of inadequate sleep. Additionally, parents identified the determinant ‘family sleep habits’, including not having or deviating from a bedtime routine, indistinctness about the child’s bedtime, and parental absence when the child needs attention. This finding aligns with the conclusions of previous reviews, and confirms parents’ critical role in children’s sleep hygiene [16,17,29]. Consequently, future sleep interventions may include enhancing parent’s self-efficacy and sleep-related parenting skills, e.g., to create and adhere to a consistent sleep schedule and a relaxing bedtime routine. Parents’ self-efficacy may be increased by using a method such as self-monitoring of behaviour, for which parents log the sleep schedule and bedtime routine activities of their child and subsequently receive feedback on these logs from a health professional [15].

Four perceived behavioural determinants were identified; energy, physical activity, activating activities, and dietary behaviour. It is generally accepted that adequate physical activity during the day promotes sleep at night [30,31]. This presumable relationship seems to be bidirectional, as the quantity and quality of sleep also seems related to physical activity the following day [32,33], creating a vicious circle [31]. Sleep is also related to diet [32,34]. Although most studies investigate inadequate sleep as a determinant of unhealthy dietary behaviours, the evidence for the bi-directionality of this relationship is growing [35]. In addition, a recent review among healthy adults [36] found that sleep was only impaired after vigorous intensity exercise, ending ≤ 1 h before bedtime. Furthermore, stimulating activities, including screen behaviours, are also generally acknowledged as important determinants of children’s inadequate sleep [37]. Such activities increase alertness, through several mechanisms: sleep time displacement, psychological stimulation and light exposure [38]. Encouraging sensible screen behaviours before bedtime (e.g., no agitating screen-based activities, avoiding screens in the bedroom) seems a promising element of future sleep interventions, but must be combined with other sleep hygiene strategies, e.g., regular bedtimes and a bedtime routine [38]. To conclude, these interactions and bidirectional relationships show that healthy sleep and its determinants should be included in healthy lifestyle interventions for children. To date, these interventions mainly focused on promoting a healthy diet and adequate amounts of physical activity [39].

Shortcomings in children’s sleep environment, including too much light, not the right temperature, noise, uncomfortable sleeping place, uncomfortable sleep materials, and distractions in the bedroom, were identified as potential determinants. This finding is partly in line with Bagley et al. who found that temperature and noise were associated with more sleep-wake problems in 10–13 year olds, while the amount of light was not related to any of their sleep outcomes [24]. The review by Allen et al. also showed limited support for adjusting the bedroom light to as dark (i.e., only two studies) and as quiet (i.e., only two studies) as possible for better sleep [17]. In contrast, a recent review of the WHO found evidence for a relationship between ambient noise and inadequate sleep [40]. Furthermore, a previous study (2017) where 11–12 year old multi-ethnic adolescents identified sleep-disturbing household activities, found that disorganization in the home environment such as TV or noise disturbance, family members phone calling, or night-time home visitors, were related to disturbed sleep [41]. In conclusion, environmental factors can be sleep-disrupting for some children, depending on children’s individual preferences for the amount of light, room temperature, noise level, and other distractions.

### 4.1. Strengths and Limitations

A strength of this study is that it provides valuable and new information about potential determinants of children’s inadequate sleep as seen from the perspective of children and their parents. These perspectives are relevant for future healthy sleep interventions. Moreover, this is one of the first studies that specifically focuses on families living in low SEP neighbourhoods. The potential determinants identified in this study may also be relevant for children living in middle and high SEP areas, however, future research is needed to confirm this. The broad focus on all aspects of inadequate sleep (duration, quality) further strengthens this study. However, there were also some limitations. Firstly, the concept mapping method makes use of a focus statement, which forces the researcher to choose one side of the health behaviour (i.e., positive or negative). This study focused on inadequate sleep, which inevitably neglects specific facilitators of healthy sleep. Secondly, potential personal determinants of which participants are unaware, e.g., unhelpful beliefs, may be overlooked. This may be included in future studies by incorporating follow-up questions focused on beliefs in the brainstorm session. Thirdly, it was not always clear why the children grouped certain ideas together. To give an example: a child (group 4A) clustered ‘In bed, thinking about something scary that I experience’ together with ‘A brother or sister that keeps me awake’. We therefore recommend that future concept mapping studies incorporate time to ask children to explain their clusters. Lastly, we did not screen the participating children and parents on potential sleep problems or other medical conditions. There is a possibility that poor sleepers and parents whose children have trouble sleeping were more interested in participating in this study despite the emphasis on recruiting children and parents with both adequate and inadequate sleep health.

Overall, all determinants were rated lower on importance by children than parents. A possible explanation may be found in the use of a smiley face Likert scale, as children tend to choose happy smiley faces rather than unhappy and very unhappy faces [42]. For future studies, we therefore recommend using smiley scales with only positive responses, varying in degree of happiness [42]. Although children’s importance ratings were lower in general, they still provided insight into the relative importance of determinants according to children.

### 4.2. Implications for Research and Practice

Our study identified potential determinants of children’s inadequate sleep health, of which the majority have not been studied thoroughly. Future research is therefore needed to confirm whether these are actual determinants of children’s inadequate sleep. In studying determinants of sleep, conducting a longitudinal study is not always required, as many psychological and social environmental determinants have acute effects. A suitable method would be ecological momentary assessment, for which potential determinants (e.g., affective state) are measured before bedtime and linked to subsequent sleep [43]. 

This study provides a broad overview of the perspectives of children and parents on potential determinants of children’s inadequate sleep. Our findings suggest that children’s sleep is affected by multiple interrelated and interacting determinants within the personal (i.e., physical, psychological and behavioural), social- and physical environment, which is in line with the social ecological model [15] and the ‘determinants of health’ model of Dahlgren and Whitehead [44]. This implies that a multilevel and multifactorial intervention is recommended to promote healthy sleep [45]. 

The results of our concept mapping study show that children’s inadequate sleep is a complex problem, with many different and interacting determinants on several levels of the social ecological model [15]. Such complex problems demand a systems approach [46,47], where relevant determinants of children’s sleep health are targeted at multiple levels of the system. Furthermore, children and parents identified many potential determinants in the current study. However, the relevance of these determinants might differ per child. For the development of a relevant healthy sleep intervention, we therefore recommend working closely with children and parents. 

## 5. Conclusions

Both children and parents identified various potential psychological (i.e., fear, affective state, stressful situation), social environmental (i.e., sleep schedule, family sleep habits), behavioural (i.e., screen behaviour, physical activity, diet), physical environmental (i.e., sleep environment), and physiological (i.e., physical well-being) determinants of children’s inadequate sleep health. These findings indicate that children and parents perceive children’s sleep health to be influenced by multiple determinants at different levels, and that children’s sleep health requires promotion through systemic, multilevel and multifactorial action. 

## Figures and Tables

**Figure 1 ijerph-17-01583-f001:**
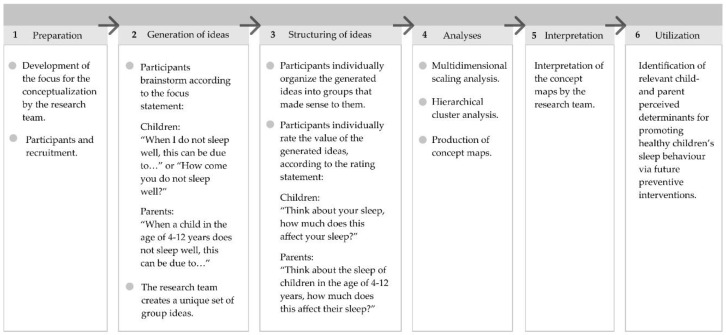
The 6-step concept mapping process.

**Table 1 ijerph-17-01583-t001:** Sample characteristics.

Characteristics		
**Children (*N* = 45)**		
	Age, years (M, SD)	10.2 (1.1)
	Female (%)	62.2
	Born in the Netherlands (%)	88.9
**Parents (*N* = 33)**		
	Age, years (M, SD)	40.4 (7.9)
	Female (%)	90.9
	Education level (%)	
	Low	9.1
	Medium	42.4
	High	42.4
	Unknown	6.1
	Perceived cultural group (%)	
	Dutch	36.4
	Turkish	15.2
	Moroccan	6.1
	Ghanaian	3.0
	Other	12.1
	Two or more groups	27.3

*N* = number of participants; M = Mean; SD = Standard deviation.

**Table 2 ijerph-17-01583-t002:** Mean importance ratings ^1^ for the child-perceived determinants related to children’s inadequate sleep.

Perceived Determinants	Main Ideas (Merged)	Examples of Underlying Original Ideas	Mean Rating per Child Group ^2^						Mean ^3^
			*1*	*2*	*3A*	*3B*	*4A*	*4B*	
***Psychological determinants***									
**Fear**									*2.9*
	Recent scary event	*‘In bed, thinking about something scary that I experienced’*	N.A.	**3.6**	N.A.	N.A.	**3.1**	**3.3**	**3.3**
	Scary thoughts	*‘Having scary thoughts when I am in bed’*	**3.3**	**3.6**	2.9	N.A.	N.A.	N.A.	**3.3**
	Nightmares	*‘Having a nightmare’*	**3.1**	**3.2**	**3.2**	2.8	**3.0**	**3.2**	**3.1**
	Watch/read something scary	*’Having seen a scary movie’, ‘Reading a moving story before I go to sleep’*	2.7	**3.0**	N.A.	2.6	**3.1**	2.7	2.8
	Scared by something in the bedroom	*‘When I see scary shadows in my bedroom’*	1.4	2.2	**3.4**	N.A.	N.A.	**3.3**	2.6
	Being afraid	*‘Being afraid when I am in bed’*	2.4	N.A.	2.8	2.4	2.1	**3.2**	2.6
	Scary sounds	*‘Hearing weird or scary sounds during the night’*	N.A.	N.A.	2.6	2.4	N.A.	N.A.	2.5
**Affective state**									*2.6*
	Many thoughts	*‘Thinking and having many thoughts’*	N.A.	N.A.	**3.8**	**4.3**	N.A.	2.8	**3.6**
	Upcoming stressful event	*‘Being nervous for something that is going to happen’*	**3.3**	**3.0**	N.A.	2.9	**3.5**	2.6	**3.1**
	Excitement	*‘Looking forward to something that will happen the next day’*	2.6	**3.0**	**3.0**	**3.1**	**3.3**	2.8	2.9
	Negative affective state	*‘Feeling sad’, ‘Being angry’, ‘Being irritated’*	2.4	2.9	2.8	2.5	**3.0**	**3.3**	2.8
	Reluctant to go to sleep	*‘Not feeling like going to sleep’*	2.4	N.A.	2.7	2.4	**3.0**	**3.6**	2.8
	Recent stressful event	*’Continuing to think about a bothersome event that happened that day’*	2.3	2.5	**3.0**	2.5	2.2	2.9	2.6
	Stressful family situation	*‘A fight between my parents when I am in bed’*	N.A.	N.A.	N.A.	2.2	3.4	1.7	2.4
	Feeling unsafe	*‘Not feeling comfortable because of people screaming outside’*	N.A.	N.A.	N.A.	2.3	2.2	2.1	2.2
	Fear of missing out	*‘When my brother/sister is allowed to watch something (TV, film) and I am not’*	N.A.	N.A.	2.0	N.A.	N.A.	N.A.	2.0
	Lacks attention from parents	*‘When my parents do not pay attention to me because they are busy with my brother or sister’*	N.A.	N.A.	2.0	N.A.	N.A.	N.A.	2.0
*Physiological determinants*									
**Discomfort**									*2.8*
	Illness	*‘Being ill’, ‘Having a blocked nose due to a cold’*	**3.1**	**3.4**	2.9	**3.5**	**3.9**	**3.2**	**3.3**
	Pain	*‘Feeling pain’*	2.9	**3.0**	N.A.	2.5	**3.6**	2.2	2.8
	Needing to pee	*‘Needing to pee when I am already in bed’*	2.8	N.A.	N.A.	2.6	2.7	2.4	2.6
	Unhealthy dietary behaviour	*‘Having had too much to eat’, ‘Late dinner’, ‘Feeling hungry’*	N.A.	N.A.	N.A.	2.4	2.7	2.6	2.6
*Physical environmental determinants*									
**Sleep environment**									*2.7*
	Not the right temperature	*‘Feeling too hot or too cold when I am in bed’*	**3.8**	N.A.	**3.0**	2.8	**3.6**	2.9	**3.2**
	Unable to lie down comfortably	*‘Not able to lie down comfortably in my bed’*	**3.8**	N.A.	**3.2**	2.4	N.A.	**3.2**	**3.1**
	Noise outside	*‘Noise from the neighbours, e.g., yelling or music’*	**3.0**	**3.3**	N.A.	2.4	2.9	**3.1**	2.9
	Uncomfortable sleeping materials	*‘Having no comfortable pillow’, ‘Wearing uncomfortable pyjamas’*	2.9	N.A.	**3.2**	N.A.	N.A.	2.7	2.9
	Uncomfortable bed	*‘Sleeping on an uncomfortable mattress’*	N.A.	N.A.	N.A.	N.A.	N.A.	2.9	2.9
	Too much light	*‘Too much light in my bedroom’*	2.9	**3.0**	N.A.	2.4	N.A.	2.8	2.8
	Noise inside	*‘When the sound of the TV in our home is too loud’*	2.8	2.2	2.5	2.4	2.9	2.59	2.6
	Distractions in the bedroom	*‘Pets that wake me up’, ‘Noise from a brother or sister with whom I share the same room’*	2.4	2.4	2.1	2.0	2.5	2.2	2.3
	Unfavourable sleeping place	*‘Not sleeping at my favourite sleeping place in the bunk bed’*	2.1	N.A.	N.A.	N.A.	N.A.	N.A.	2.1
	Too dark	*‘A bedroom that is too dark’*	N.A.	1.8	N.A.	N.A.	N.A.	N.A.	1.8
*Social environmental determinants*									
**Sleep schedule**									*2.6*
	Going to bed too early	*‘Going to bed too early and not being tired enough to fall asleep’*	2.8	N.A.	**3.2**	**3.6**	**3.3**	N.A.	**3.2**
	No consistent sleep schedule	*‘Not going to bed at the same time every night’*	N.A.	N.A.	2.6	2.0	1.9	2.2	2.2
	Daytime nap	*‘Sleeping at daytime’*	N.A.	N.A.	N.A.	N.A.	2.7	2.0	2.4
*Behavioural determinants*									
**Energy**									*2.4*
	Not being tired	*‘Not being tired when I go to bed’*	**3.9**	N.A.	N.A.	N.A.	**3.1**	N.A.	**3.5**
	Inadequate amount of daytime PA	*‘Having had too little exercise during the day and therefore too much energy in the evening’*	2.1	N.A.	N.A.	2.5	N.A.	N.A.	2.3
	Excessive daytime stimulation	*‘Being too tired because I did a lot’*	N.A.	N.A.	N.A.	1.9	N.A.	2.22	2.1
	Evening PA	*‘Doing sports late in the evening’*	1.5	N.A.	2.4	N.A.	1.9	N.A.	1.9
**Screen behaviour**									1.9
	Screen use around bedtime	*‘Using a phone or tablet before bedtime’*	2.5	2.0	N.A.	2.1	1.9	1.9	2.1
	Social media use around bedtime	*‘Messages I receive in the evening* via *the group chat on my phone’*	1.7	2.0	N.A.	N.A.	N.A.	N.A.	1.9
	Playing activating games before bedtime	*‘Playing computer games right before going to sleep’*	1.4	N.A.	N.A.	2.0	N.A.	N.A.	1.7

N.A. = Not applicable, means the idea was not mentioned in this group of children; bold values indicate the determinant or idea was perceived as important i.e., ≥3.00; cursive values indicate the overall mean rating per perceived determinant. PA = Physical activity. ^1^ The mean importance rating is based on the question: ‘Think about your sleep, how much does this affect your sleep?’ answered on a 5-point Likert scale from ‘does not affect at all = 1’ to ‘affects a whole lot’ = 5’. ^2^ Groups of children per school (i.e., schools 1–4). At schools 3 and 4, there were two groups of children, represented by A and B. ^3^ Mean importance rating of all groups of children.

**Table 3 ijerph-17-01583-t003:** Mean importance ratings ^1^ for the parent-perceived determinants related to children’s inadequate sleep.

Perceived Determinants	Main Ideas (Merged)	Examples of Underlying Original Ideas	Mean Rating per Parent Group ^2^				Mean ^3^
			*1*	*2*	*3*	*4*	
***Physiological determinants***							
**Physical well-being**							***3.9***
	Illness	*‘Being ill’*	**4.1**	**4.2**	**4.2**	**3.5**	**4.0**
	Pain	*‘Feeling pain’*	N.A.	N.A.	**4.2**	N.A.	**4.2**
	Sleep problem	*‘Sleep walking’, ‘Wetting the bed’*	**3.0**	N.A.	N.A.	**4.0**	**3.5**
*Psychological determinants*							
**Stressful situation**							***3.8***
	Feeling unsafe	*‘Not feeling safe at home, in the classroom or outside on the streets’*	N.A.	N.A.	**4.1**	**4.7**	**4.4**
	Parental relationship problems	*‘Negative tension or disagreement within the family’, ‘Parents going through a divorce’*	**3.4**	**3.8**	**4.5**	**4.4**	**4.0**
	Being bullied		**3.4**	**3.8**	**4.2**	**4.7**	**4.0**
	Insecurity about themselves	*‘Feeling insecure about themselves’*	**4.0**	N.A.	N.A.	N.A.	**4.0**
	Parental stress	*‘Parental stress (rushing) that is transferred to the child’*	**3.9**	N.A.	**3.1**	**4.3**	**3.8**
	Financial family problems	*‘Financial problems at home, meaning the parent is unable to buy everything for the child’*	N.A.	2.6	N.A.	N.A.	2.6
**Affective state**							***3.6***
	Unpleasant dreams	*‘Nightmares, ‘Dreams that keep children awake or wake them and cause restless sleep’*	N.A.	N.A.	**3.3**	**4.5**	**3.9**
	Upcoming stressful event	*‘A stressful event coming up for the child the next day’*	**4.0**	**3.8**	**3.6**	**3.5**	**3.7**
	Being afraid	*‘Being afraid when lying in bed’*	N.A.	**3.6**	3.1	**4.5**	**3.7**
	Many thoughts	*‘Having many thoughts’*	N.A.	N.A.	**3.3**	**4.2**	**3.7**
	Recent stressful event	*‘Continuing to think about something that happened that day’*	**3.9**	2.9	**3.6**	**4.3**	**3.7**
	Excitement	*‘Excitement, happy feelings for something that is going to happen the next day’*	**3.6**	**3.6**	N.A.	**3.2**	**3.4**
	Fear of missing out	*‘Not willing to miss something and therefore not willing to go to sleep’*	N.A.	**3.9**	**3.1**	**3.2**	**3.4**
	Worrying	*‘Worrying about something and not being able to share this’*	**3.6**	2.9	**3.7**	N.A.	**3.4**
	Reluctant to go to sleep	*‘Not feeling like going to sleep’*	**3.0**	N.A.	N.A.	**3.3**	**3.2**
*Behavioural determinants*							
**Energy**							***3.5***
	Being too energetic	*‘Not being tired when going to bed’, ‘Having too much energy from his/herself’*	**3.9**	N.A.	N.A.	**3.8**	**3.8**
	Excessive daytime stimulation	*‘A busy day with excessive stimulation due to too many activities’*	N.A.	N.A.	**3.4**	**4.0**	**3.7**
	Inadequate daytime stimulation	*‘A boring day with inadequate stimulation due to lack of activities’*	**3.6**	N.A.	**3.1**	**3.2**	**3.3**
	Being too tired	*‘Being too tired when going to bed’*	**3.3**	N.A.	**3.6**	2.8	**3.2**
**Activating activities**							**3.4**
	Watching something scary	*‘Watching a scary movie’, ‘Watching the news’*	**3.4**	**4.0**	**3.6**	**3.8**	**3.7**
	Play with activating toys before bedtime	*‘Playing with toys with a lot of light and noise right before bedtime’*	N.A.	**3.1**	**4.4**	N.A.	**3.8**
	Screen use before bedtime	*‘Using the computer or other screen (phone, tablet, game computer, TV) right before going to sleep’*	**3.4**	**3.3**	**3.7**	**3.3**	**3.4**
	Excessive daytime screen use	*‘Using screens (phone, tablet, game computer, TV) a lot during the day’*	N.A.	N.A.	N.A.	2.7	2.7
**Physical activity**							***3.3***
	Inadequate amount of daytime PA	*‘Inadequate amount of physical activity during the day and therefore not being tired’*	N.A.	**3.6**	N.A.	**3.2**	**3.4**
	Inadequate time outside at daytime	*‘Not having played outside at daytime’, ‘Spending too little time outside in the fresh air’*	**3.0**	N.A.	N.A.	**3.3**	**3.2**
**Diet**							***3.1***
	Unhealthy diet	*‘Eating something sugary before bedtime’, ‘Unhealthy diet during the day’*	**3.4**	**3.6**	N.A.	2.5	**3.2**
	Excessive amount of food close to bedtime	*‘Eating too much right before going to bed’*	**3.3**	**3.5**	N.A.	2.7	**3.2**
	Drinking too much before bedtime	*‘Drinking too much before going to sleep and therefore needing to go to the toilet often’*	N.A.	N.A.	N.A.	**3.2**	**3.2**
	Did not drink enough	*‘Feeling thirsty’, ‘Lack of water during the day’*	N.A.	N.A.	N.A.	**3.0**	**3.0**
	Inadequate amount of food	*‘Feeling hungry during the night’*	2.6	N.A.	N.A.	**3.2**	2.9
*Social environmental determinants*							
**Sleep schedule**							***3.5***
	Too early bedtime	*‘A bedtime that is too early for child’s circadian rhythm’*	N.A.	N.A.	N.A.	**3.8**	**3.8**
	No consistent sleep schedule	*‘No consistent sleep times’, ‘Irregular sleep times during weekends’*	**3.9**	**3.5**	**3.4**	2.8	**3.4**
	Daytime nap	*‘Napping in the afternoon’*	N.A.	**3.0**	N.A.	**3.5**	**3.3**
**Family sleep habits**							**3.3**
	No bedtime routine	*‘Having no bedtime routine’*	N.A.	**3.3**	N.A.	**3.7**	**3.5**
	Parental absence	*‘Absence of the mother or father when the child needs attention’*	N.A.	**4.0**	N.A.	2.9	**3.5**
	Indistinctness about bedtime	*‘Not indicating clearly when the child needs to go to bed’*	N.A.	**3.2**	N.A.	N.A.	**3.2**
	Deviate from bedtime routine	*‘When the parent deviates from the usual bedtime routine’*	2.6	2.7	**3.4**	**3.5**	**3.1**
**Social norms**							*2.9*
	Social bedtime norm among siblings	*‘Older brothers or sisters that are allowed to stay up longer’*	N.A.	**4.2**	2.7	2.8	**3.2**
	Social bedtime norm among classmates	*‘Other children in their class that are allowed to stay up longer’*	N.A.	2.8	2.3	N.A.	2.6
*Physical environmental determinants*							
**Sleep environment**							***3.0***
	Seasonal changes	*‘When it is still light outside when they need to go to bed’*	2.9	**4.0**	N.A.	**3.2**	**3.4**
	Absence of favourite sleep accessory	*‘The absence of their favourite stuffed animal or sleeping cloth’*	N.A.	2.7	**3.1**	**4.2**	**3.3**
	Noise outside	*‘Fighting neighbours’, ‘Noise from the street’*	N.A.	**3.6**	2.9	**3.5**	**3.3**
	Not the right temperature	*‘A bedroom that is too hot or too cold’*	**3.1**	**3.4**	N.A.	**3.1**	**3.2**
	Reluctant to sleep alone	*‘Wanting to stay with their parent and not wanting to be alone’*	**3.3**	N.A.	**3.2**	**3.0**	**3.2**
	Uncomfortable sleeping material	*‘Uncomfortable sleeping attributes, such as pillows, pyjamas, underwear’*	N.A.	**3.1**	N.A.	N.A.	**3.1**
	Too much light	*‘Too much light in the bedroom’*	2.7	**3.4**	N.A.	**3.0**	**3.1**
	Noise inside	*‘Too much noise within the home’*	N.A.	N.A.	**3.0**	**3.0**	**3.0**
	Different sleep environment	*‘A different environment, not sleeping in their own bed’*	N.A.	**3.0**	N.A.	**3.0**	**3.0**
	Uncomfortable place to sleep	*‘Not having a comfortable bed and therefore not being able to lay down comfortably’*	N.A.	**3.0**	N.A.	**3.0**	**3.0**
	No fresh air	*‘No fresh air in the bedroom’*	N.A.	**3.0**	N.A.	N.A.	**3.0**
	Distractions in the bedroom	*‘A less tidy and messy bedroom’, ‘Shared bedroom with brothers or sisters’*	2.8	**3.0**	N.A.	**3.2**	2.9
	Too dark	*‘A bedroom that is too dark’*	2.9	2.9	2.9	N.A.	2.9
	Too quiet	*‘When it is too quiet at home’*	N.A.	N.A.	2.1	N.A.	2.1

N.A. = Not applicable, means the idea was not mentioned in this group of parents; bold values indicate that the determinant or idea was perceived as important i.e., ≥3.00; cursive values indicate the overall mean rating per perceived determinant; PA = Physical activity. ^1^ The average importance ratings were based on the question: ‘Think about the sleep of a child in the age of 4–12 years, how much does this affect their sleep?’ answered on a 5-point Likert scale from ‘does not affect at all = 1’ to ‘affects a whole lot = 5’. ^2^ Groups of parents per school (i.e., schools 1–4). ^3^ Mean importance rating of all groups of parents.

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
