# Peer review of "Child and Parent Perceived Determinants of Children’s Inadequate Sleep Health. A Concept Mapping Study"

_ijerph, 2020, doi:10.3390/ijerph17051583_

Round 1
Reviewer 1 Report
The conclusions reached by the authors based on the results obtained are appropriate, but perhaps it would be interesting to state possible ways of acting to achieve the determintates that make the sleep of the children surveyed healthier
Author Response
We thank the reviewer for his/her time to review our manuscript and the valuable feedback. We revised the manuscript and integrated the reviewers’ feedback. Below we provide a point-to-point reply to the Reviewers’ comments.
Comment 1.1
The conclusions reached by the authors based on the results obtained are appropriate, but perhaps it would be interesting to state possible ways of acting to achieve the determintates that make the sleep of the children surveyed healthier.
Response 1.1.
We thank the author for the compliment of noting that the conclusions of our study validly follow our results and the suggestion to further improve our manuscript. We agree with the reviewer that it is interesting to elaborate a bit more on how positive changes in certain determinants may be achieved and therefore we included this in the discussion of our revised manuscript.
This now reads...
“Concept mapping is a 6-step process (see Figure 1). The first five steps are illustrated below. Step six included the identification of perceived determinants relevant for including in future healthy sleep interventions (see Discussion).” [Methods, line 73 - 77] “This may be included in healthy sleep interventions by teaching children relaxation techniques, such as meditation, breathing exercises, imagination journeys [27] in both school, after-school and community settings. Additionally, teaching parents (i.e. through online or community program) on how to implement such techniques in their child’s bedtime routine could be an important avenue to promote healthy sleep.” [Discussion, line 264 - 269] “Our findings support existing evidence that creating a positive and supportive family environment is important for healthy sleep. It may therefore be valuable to encourage parents to monitor their child’s needs, desires and stressors. Such intervention may include letting parents re-evaluate their behaviour, giving scenario-based risk information, or raising consciousness to increase parents’ awareness [15]. Increasing awareness must be quickly followed by increasing parent’s problem solving ability and self-efficacy (i.e. confidence in their ability) by using methods such as goal setting, self-monitoring of their behaviour, setting graded tasks, and planning coping responses [15]. This may reach parents via face-to-face or web-based sessions.” [Discussion, line 280 - 289] “Thus, future sleep interventions may include enhancing parent’s self-efficacy and sleep-related parenting skills, e.g., to create and adhere to a consistent sleep schedule and a relaxing bedtime routine. Parents’ self-efficacy may be increased by using a method such as self-monitoring of behaviour, where parents log the sleep schedule and bedtime routine activities of their child followed by feedback on these logs by a health professional [15].” [Discussion, line 296 - 301] “Sensible screen behaviours before bedtime (e.g. no arousing screen-based activities, avoiding screens in the bedroom) seems a promising element for future sleep interventions, but must be combined with other sleep hygiene strategies, e.g., regular bedtimes and a bedtime routine [39].” [Discussion, line 313 - 315]
Reviewer 2 Report
Environmental Research Review
Into-
There was some related research:
https://www.ncbi.nlm.nih.gov/pubmed/19097573
1/38- please reference newest NSF guidelines for age-specific sleep recommendations
Methods-
-It would have been preferable to have administered some sleep screener to the sample. It is difficult to know if there was a bias in those who agreed to participate re: having a higher prevalence of sleep problems than the population.
-How if at all were children grouped? Regarding the above, I would have liked to see some type of grouping or analysis by child’s perceived sleep quality and duration.
-Do we know if the children had any health conditions that could impact their percpetivies?
-I
2/68-77- As someone unfamiliar with the mapping approach please: provide an over view of the approach, how it has been used similarly, and define terms (e.g., focus statement, rating statement).
3/106- what was the process for merging ideas’?
159- please define education categories for an international audience
Figures in the appendix were difficult to interpret.
Author Response
We thank the reviewer for his/her time to review our manuscript and the valuable feedback. We revised the manuscript and integrated the reviewers’ feedback. Below we provide a point-to-point reply to the Reviewers’ comments.
Comment 2.1
English language and style are fine/minor spell check required.
Response 2.1
We double checked the full manuscript text and appendixes for any language issues and made some changes accordingly. We tracked all spelling changes in the manuscript.
Comment 2.2
There was some related research: https://www.ncbi.nlm.nih.gov/pubmed/19097573 (introduction)
Response 2.2
We thank the reviewer for the literature suggestion of Holmberg and Hellberg (2008). However, we believe that comparing our results with this study would be inappropriate, as they focused on a different age group (adolescents aged 13-18 years) and the determinants in their study were selected by researchers. This is different from our study wherein we focused on 4-12 year old children and wherein the children and parents selected the determinants.
Comment 2.3
1/38- please reference newest NSF guidelines for age-specific sleep recommendations (introduction).
Response 2.3
Thank you for this suggestion, we added the newest NSF guidelines to this version of the manuscript.
This now reads…
“For children, healthy sleep is typically operationalized as a regular sleep rhythm consisting of approximately 9-11 hours per night of good quality sleep (i.e. the combination of high sleep efficiency and a good subjective assessment of their own sleep) [1, 2].” [Introduction, line 42-45]
“2. Hirshkowitz, M.; Whiton, K.; Albert, S. M.; Alessi, C.; Bruni, O.; DonCarlos, L.; Hazen, N.; Herman, J.; Adams Hillard, P. J.; Katz, E. S.; Kheirandish-Gozal, L.; Neubauer, D. N.; O'Donnell, A. E.; Ohayon, M.; Peever, J.; Rawding, R.; Sachdeva, R. C.; Setters, B.; Vitiello, M. V.; Ware, J. C., National Sleep Foundation's updated sleep duration recommendations: final report. Sleep health 2015, 1, (4), 233-243.”
[References, line 599 - 602]
Comment 2.4
It would have been preferable to have administered some sleep screener to the sample. It is difficult to know if there was a bias in those who agreed to participate re: having a higher prevalence of sleep problems than the population. (methods) Do we know if the children had any health conditions that could impact their perspectives? (methods)
Response 2.4
We agree with the reviewer that it would be preferable to have more information on our sample such as potential sleep problems or other medical conditions. Since we do not have this information, we added this to the limitation section of the discussion.
This now reads…
“Last, we did not screen the participating children and parents on potential sleep problems or other medical conditions. There is a possibility that poor sleepers and parents whose children have trouble sleeping were more interested in participating in this study despite the emphasis on recruiting children and parents with adequate as well as inadequate sleep health.” [Discussion, line 349 – 353].
Comment 2.5
How if at all were children grouped? Regarding the above, I would have liked to see some type of grouping or analysis by child’s perceived sleep quality and duration. (methods)
Response 2.5
We thank the reviewer for raising this point. Children and parents were recruited through schools and thereby grouped based on school level and their availability. Moreover, in concept mapping, data are collected on a group level and not an individual level. Participants were also prompted to think of the experience of other children of their age (for children) or of other children aged 4-12 years (for parents) rather than only their individual experience. We made some changes accordingly.
This now reads…
“Children and parents were recruited through schools and thereby grouped based on school level and their availability. The health advisors of the Public Health Service of Amsterdam brought the researchers in contact with primary schools in socioeconomically disadvantaged neighbourhoods, based on the postal code of the neighbourhood [21]. Four out of 23 invited primary schools participated.”[Methods, line 96 – 100]. “A participatory mixed-methods concept mapping study was conducted to assess children’s and parents’ perspectives on potential determinants of children’s inadequate sleep health [18]. The qualitative part of this approach included generating ideas on potential determinants from participants on group level via a brainstorm session and sorting these ideas according to relatedness; followed by rating the ideas on their importance.” [Methods, line 68 – 73].
Comment 2.6
2/68-77- As someone unfamiliar with the mapping approach please: provide an over view of the approach, how it has been used similarly, and define terms (e.g., focus statement, rating statement). (methods)
Response 2.6
Based on the reviewers suggestions we have further elaborated on the concept mapping approach, the defined terms and how it has been used similarly.
This now reads…
“First, a ‘focus statement’ and ‘rating statement’ were created: a main question or statement that gives a specific instruction for the session [18]. The purpose of the focus statement is to elicit ideas about the topic of interest, whereas the rating statement provides comparative ratings of importance for the generated ideas (see Figure 1).” [Methods, line 84 – 87]. “A participatory mixed-methods concept mapping study was conducted to assess children’s and parents’ perspectives on potential determinants of children’s inadequate sleep health [18]. The qualitative part of this approach included generating ideas on potential determinants from participants on group level via a brainstorm session and sorting these ideas according to relatedness; followed by rating the ideas on their importance. Researchers are not allowed to add or prompt additional ideas. Concept mapping is a 6-step process (see Figure 1). The first five steps are illustrated below. Step six included the identification of perceived determinants relevant for including in future healthy sleep interventions (see Discussion). Concept mapping has been used in previous studies among children and perspectives on behavioural determinants [19, 20]. Additional information about the concept mapping approach can be found elsewhere [18].” [Methods, line 68 – 79].
Comment 2.7
3/106- what was the process for merging ideas’? (methods).
Response 2.7
After each brainstorm session, two researchers (LSB and IAH) merged conceptually similar ideas resulting in a list of unique ideas. We elaborated upon this process in the manuscript.
This now reads…
“After each brainstorm session, ideas that were conceptually the same were merged, resulting in a set of unique ideas which were printed on paper cards for the second session. One researcher (L.S.B.) suggested the adaptations, which were checked by a second researcher (I.A.H.). In case of disagreement, a third researcher (V.B. or M.M.v.S.) was consulted.” [Methods, line 124 – 127].
Comment 2.8
159- please define education categories for an international audience (methods).
Response 2.8
We added a definition of the education categories to the methods section of the manuscript.
This now reads…
“During the first session, participants also completed a short questionnaire including age, gender, education level (parents), and perceived cultural group (parents) or country of birth (children). Parents’ education categories were defined as 1) low, i.e. highest education level is primary or middle school education or no education at all; 2) medium, i.e. highest education is secondary vocational education; 3) high, i.e. highest education is higher professional education or scientific education.” [Methods, line 119 - 124].
Comment 2.9
Figures in the appendix were difficult to interpret.
Response 2.9
We clarified the descriptions of all our Figures in the appendix. We hope the descriptions are clear now.
This now reads…
“Figure [nr.]. Concept map [children/parents] group [group nr.]. In this map, each point reflects one idea. Ideas that were sorted together more often by the participants appear closer to each other on the map. Ideas never/rarely sorted together appear widely separated on the map. Clusters are groups of ideas that were sorted together most often and reflect conceptually related ideas according to the participants in this group.” [Appendix A, line 417 - 511]. “The defined cluster names in this concept map were: Cluster 1: …” [Appendix A, line 417 - 511].
Reviewer 3 Report
"Child- and parent perceived determinants of children’s inadequate sleep health. A concept mapping study"
The paper describes a concept-mapping study which investigates perspectives of children and parents on potential determinants of children’s inadequate sleep. Overall, the description of the study is clear and discusses an important topic. I have a few comments, as follows.
I think that the claim of novelty is overstated. I did not see any surprising or unknown findings resulting from the study. The fact that being bullied, having parental stress at home, etc., disrupts children’s sleep is hardly new. Similarly, “may therefore be valuable to increase parents’ awareness of the relevance to monitor their child’s needs, desires and stressors closely when something happens within their family environment” is nothing new. The study shows evidence that supports existing, well-known determinants, and I think the statement of the paper’s contributions should reflect that. A main focus of the study is how potential determinants are affected by or at least related to low socioeconomic status. A major drawback of the paper is that it provides no comparison. The study was conducted only in underprivileged areas, so it is impossible to say how the results are special to this cohort. I think they are not. Even affluent children who suffer from bullying and parental stress (for example) would lose sleep. Same thing with nightmares, and illness – they, too, affect children’s sleep regardless of socioeconomic status. It seems that there was limited value to soliciting statements from the children, as in the end, mostly parents’ statements were useful for the concept mapping study. The concept mapping analysis could have benefited from investigating additional predictors. For example, demographic aspects such as sex, number of siblings, etc. I feel a little uneasy about tweaking the clusters in the concept mapping manually (section 2.2). In addition to the decisions which were provided in an Appendix, it may be good to explain how these decisions changed the conclusions. Too much intervention on part of the researcher detracts from the relevance of the study, as the concepts arising from the study may not reflect the joint sentiment that the concept mapping methodology is designed to find.Author Response
We thank the reviewer for his/her time to review our manuscript and the valuable feedback. We revised the manuscript and integrated the reviewers’ feedback. Below we provide a point-to-point reply to the Reviewers’ comments.
Comment 3.1
English language and style are fine/minor spell check required
Response 3.1
We double checked the full manuscript text and appendices and made some changes accordingly. We tracked all spelling changes in the manuscript.
Comment 3.2
The paper describes a concept-mapping study which investigates perspectives of children and parents on potential determinants of children’s inadequate sleep. Overall, the description of the study is clear and discusses an important topic. I have a few comments, as follows.
Response 3.2.
We thank the reviewer for the compliments and for taking the time to address points of feedback to further improve our manuscript. We have split the reviewer feedback into several comments, to be able to address each comment sufficiently and provide information regarding any adjustments.
Comment 3.3
I think that the claim of novelty is overstated. I did not see any surprising or unknown findings resulting from the study. The fact that being bullied, having parental stress at home, etc., disrupts children’s sleep is hardly new. Similarly, “may therefore be valuable to increase parents’ awareness of the relevance to monitor their child’s needs, desires and stressors closely when something happens within their family environment” is nothing new. The study shows evidence that supports existing, well-known determinants, and I think the statement of the paper’s contributions should reflect that.
Response 3.3
We thank the reviewer for this comment. Apparently the novelty of our study was misunderstood. We would like to clarify that the novelty of our study is that we explored potential determinants for children’s inadequate sleep health from the perspective of parents and children themselves. These perspectives are, as far as we know, currently lacking, and therefore a novel contribution to the literature. In addition, exploring perspectives of the target population is very relevant for future intervention development. We agree with the researcher that our results support existing evidence from studies based on the perspective of researchers or clinicians. We have explained this novelty in the introduction of our manuscript: “… the perspectives of children themselves and their parents are lacking in the current literature” and “the perspectives of children and their parents could bring about new and important insights on potential determinants of inadequate sleep, which can inform intervention development”. Based on the reviewers comment we made the following revisions.
This now reads…
“A strength of this study is that it provides valuable and novel information on potential determinants of children’s inadequate sleep from the perspectives of children themselves as well as parents. These perspectives are relevant for future healthy sleep interventions.” [Discussion, line 332 – 335]. “… A recent review (2019) [16] found inconclusive evidence for a relationship between anxiety symptoms and sleep duration, and no evidence for a relationship with sleep quality. However, the results in this review are based on only two longitudinal studies. An earlier review (2013) found existing evidence for a relationship between stress or anxiety and sleep based on a variety of study designs (e.g. experimental, longitudinal and cross-sectional) among adults [24]. Additionally, Bagley et al. (2015) found that pre-sleep worries mediated the relationship between family income and children’s sleep health based on cross-sectional data [25]. Potentially, those who experienced anxiety engaged in unhelpful pre-bedtime behaviours [26]. Both children and parents in the current study rated the perceived psychological determinants, fear and affective state, as important, and these determinants included many underlying ideas. Therefore, dealing with fear or other negative affective feelings before falling asleep, concurrent with good sleep hygiene practices, may be a promising focus when promoting healthy sleep.” [Discussion – line 253 - 264]. “… A supportive and healthy family environment is characterized by parents’ involvement in their child’s life, and a good relationship between caregivers and the child [28]. This also means that parents need to be aware when their child is going through a difficult time and support their child where needed e.g. when the child had a negative experience such as being bullied. Furthermore, one longitudinal study found evidence for parent-child physical conflict (i.e., verbal aggression such as screaming, and physical aggression such as hitting) as determinant of children’s insufficient sleep duration [29]. Our findings support existing evidence that creating a positive and supportive family environment is important for healthy sleep. It may therefore be valuable to encourage parents to monitor their child’s needs, desires and stressors.” [Discussion – line 274 -283].
Comment 3.4
A main focus of the study is how potential determinants are affected by or at least related to low socioeconomic status. A major drawback of the paper is that it provides no comparison. The study was conducted only in underprivileged areas, so it is impossible to say how the results are special to this cohort. I think they are not. Even affluent children who suffer from bullying and parental stress (for example) would lose sleep. Same thing with nightmares, and illness – they, too, affect children’s sleep regardless of socioeconomic status.
Response 3.4
We apologize for the misunderstanding about the main focus of our study which was to explore the perspectives of children and parents about the potential determinants of children’s inadequate sleep health and their perceived importance of these determinants. We were specifically interested in perspectives of children and parents living in disadvantage areas. The reason for this is that we aim to develop a preventive intervention targeting healthy sleep among children living in disadvantaged areas in Amsterdam to reduce health inequalities. We completely agree with the reviewer that many perceived determinants may also be relevant for affluent children. To address this last point, we made adjustments to the abstract and discussion section of our manuscript.
This now reads…
“… Moreover, this is one of the first studies that focused specifically on families living in low SEP neighbourhoods. The potential determinants identified in this study may also be relevant for children living in middle and high SEP areas, however, future research is needed to confirm this.” [Discussion – line 335 - 338]. “… Participants included 9-12 year-old children (n = 45) and parents (n = 33) living in neighbourhoods with a low socioeconomic position. …” [Abstract, line 25 - 27].
Comment 3.5
It seems that there was limited value to soliciting statements from the children, as in the end, mostly parents’ statements were useful for the concept mapping study.
Response 3.5
We apologize for the misunderstanding but both perspectives of children themselves and the perspectives of parents, were useful. We are happy to make adjustments if the reviewer points out what sentence may have led to this confusion.
Comment 3.6
The concept mapping analysis could have benefited from investigating additional predictors. For example, demographic aspects such as sex, number of siblings, etc.
Response 3.6
We thank the reviewer for critically thinking along. We feel like it adds to the quality of our manuscript. The concept mapping methodology is a mixed-methods approach aiming to explore perspectives of the participants. Therefore, the researchers are not allowed to raise additional ideas such as sex and number of siblings. We have clarified the concept mapping methodology in the reviews manuscript to make this more clear.
This now reads…
“A participatory mixed-methods concept mapping study was conducted to assess children’s and parents’ perspectives on potential determinants of children’s inadequate sleep health [18]. The qualitative part of this approach included generating ideas on potential determinants from participants on group level via a brainstorm session and sorting these ideas according to relatedness; followed by rating the ideas on their importance. Researchers are not allowed to add or prompt additional ideas. Concept mapping is a 6-step process (see Figure 1). …” [Methods, line 68 - 73].
Comment 3.7
I feel a little uneasy about tweaking the clusters in the concept mapping manually (section 2.2). In addition to the decisions which were provided in an Appendix, it may be good to explain how these decisions changed the conclusions. Too much intervention on part of the researcher detracts from the relevance of the study, as the concepts arising from the study may not reflect the joint sentiment that the concept mapping methodology is designed to find.
Response 3.7
We would like to emphasize that the manual changes did not change the essence or content of the cluster, but instead made clusters more coherent. Consequently, these changes did not affect the overall results or conclusion. We only manually moved a few ideas to better fitting but already existing clusters. The manual changes, did not change the essence of the cluster but instead made clusters more coherent. Consequently, they did not change the overall results. Manually adjusting clusters and moving ideas according to relatedness of ideas, is not uncommon in the concept mapping methodology. We added a reference to two previous studies that similarly used this concept mapping approach in the methods section.
This now reads…
“… Concept mapping is a 6-step process (see Figure 1). The first five steps are illustrated below. Step six included the identification of perceived determinants relevant for including in future healthy sleep interventions (see Discussion). Concept mapping has been used in previous studies among children and perspectives on behavioural determinants [19, 20]. Additional information about the concept mapping approach can be found elsewhere [18].”” [Methods, line 73 - 79]

Round 2
Reviewer 2 Report
Authors have tried to be responsive to prior comments. However, this paper needs a detailed read of every sentence, to ensure that phraseology is not awkward, and comments are well supported by appropriate cites.
I attach a file with examples. However, I will note that prior comments were for authors to review the entire paper, and not just respond to the specific comments given- which they did.
Attached file:
Line 37: The NSF citation only supports sleep duration- there is no mention of sleep rhythm. As written the Buysse & Hirshkowitz references are confounded.
Line 65: “…neighborhoods with a low socioeconomic position”- less awkward
would be to write: “in lower socio-economic neighborhoods.”
Line 66: “small housing” --less awkward would be “cramped,” “crowded” housing.
Line 68: “Health promoting interventions are only effective when targeting relevant determinants.” I take issue with the word “only”- certainly targeting relevant determinants is a necessary but not sufficient condition for an intervention to be effective. Others include: extent to which it is tailored to the age/culture/literacy level, dose, etc.
Lines 288-294: These lines clearly suggest a figure or table that would compare/contrast the parent and child findings. But there is not one.
Line 298- “these perspectives brought about important insights” BUT line 273 states “None of the child-perceived determinants were rated as important.”
Discussion- overall provided nice context. Streamlining it a bit would help the main ideas come thru better.
Child perceived barriers (ages 11-12) to sleep https://www.ncbi.nlm.nih.gov/pmc/articles/PMC5373486/
There are multiple measures of children’s self-reported sleep:
https://www.ncbi.nlm.nih.gov/pmc/articles/PMC5303893/
Figure 1- this is a poor quality figure
Author Response
We thank reviewer for his/her time to review our revisions and their valuable remarks in this second review round. We revised the manuscript and integrated the feedback . Below we provide a point-to-point reply to the Reviewer’s comments.
Comment 2.1
Authors have tried to be responsive to prior comments. However, this paper needs a detailed read of every sentence, to ensure that phraseology is not awkward, and comments are well supported by appropriate cites.
I attach a file with examples. However, I will note that prior comments were for authors to review the entire paper, and not just respond to the specific comments given- which they did.
Response 2.1
We thank the reviewer for taking the effort to provide additional comments. We understand that our manuscript needs English editing and therefore, have sent our manuscript to a language editing service for detailed corrections. In addition, we double checked if all comments are well supported by the appropriate cites. All corrections are highlighted in the manuscript using track changes.
Comment 2.2
Line 37: The NSF citation only supports sleep duration- there is no mention of sleep rhythm. As written the Buysse & Hirshkowitz references are confounded.
Response 2.2
Based on the reviewer’s comment we have changed the place of the references in the text of the manuscript. We also checked the rest of the manuscript for similar issues.
This now reads…
“For children, healthy sleep is typically defined as a regular sleep rhythm consisting of approximately 9-11 hours per night [1] of good quality sleep (i.e. the combination of high sleep efficiency and a good subjective assessment of their own sleep) [2]. “
[Introduction, line 40 – 43]
- Hirshkowitz, M.; Whiton, K.; Albert, S. M.; Alessi, C.; Bruni, O.; DonCarlos, L.; Hazen, N.; Herman, J.; Adams Hillard, P. J.; Katz, E. S.; Kheirandish-Gozal, L.; Neubauer, D. N.; O'Donnell, A. E.; Ohayon, M.; Peever, J.; Rawding, R.; Sachdeva, R. C.; Setters, B.; Vitiello, M. V.; Ware, J. C., National Sleep Foundation's updated sleep duration recommendations: final report. Sleep health 2015, 1, (4), 233-243.
- Buysse, D. J., Sleep health: can we define it? Does it matter? Sleep 2014, 37, (1), 9-17.
[References, line 650 – 654]
Comment 2.3
Line 65: “…neighborhoods with a low socioeconomic position”- less awkward
would be to write: “in lower socio-economic neighborhoods.”
Response 2.3
As suggested by the reviewer, we replaced ‘neighborhoods with a low socioeconomic position’ by ‘low socioeconomic neighborhoods’ throughout the manuscript.
Comment 2.4
Line 66: “small housing” --less awkward would be “cramped,” “crowded” housing.
Response 2.4
We replaced ‘small housing’ with ‘cramped’ housing throughout the manuscript. In addition, we asked the language editor to specifically check this terminology.
Comment 2.5
Line 68: “Health promoting interventions are only effective when targeting relevant determinants.” I take issue with the word “only”- certainly targeting relevant determinants is a necessary but not sufficient condition for an intervention to be effective. Others include: extent to which it is tailored to the age/culture/literacy level, dose, etc.
Response 2.5
We thank the reviewer for this remark. We agree with the reviewer that only targeting relevant determinants is not sufficient for interventions to be effective. We hope the rewording clarifies our intentions.
This now reads…
“Knowledge of the most relevant behavioural determinants is essential for the development of effective interventions.” [Introduction, line 53 - 54]
Comment 2.6
Lines 288-294: These lines clearly suggest a figure or table that would compare/contrast the parent and child findings. But there is not one.
Response 2.6
We would like to clarify that these lines indeed compare the results from parents and children, which are presented in the manuscript in Table 2 and Table 3. Combining these tables into one table would not improve the readability and providing another table would lead to duplication of information. We therefore chose to describe the comparison in the text. However, if the editor believes a third table would be of added value we are happy to provide one.
Comment 2.7
Line 298- “these perspectives brought about important insights” BUT line 273 states “None of the child-perceived determinants were rated as important.”
Response 2.7
Indeed, children did not rate the determinants as strongly affecting their sleep. Nevertheless, our findings provide us important insights, namely which determinants children perceive as affecting their sleep and which determinants they find most important. We made changes to emphasize this in our manuscript.
This now reads…
“None of the child-perceived determinants were rated as important (≥3.00) by the children with an average score ranging from 1.9 to 2.9.” [Results, line 225 – 226]
“Overall, all determinants were rated lower on importance by children than parents. A possible explanation may be found in the use of a smiley face Likert scale, as children tend to choose happy smiley faces rather than unhappy and very unhappy faces [42]. For future studies, we therefore recommend to using smiley scales with only positive responses, varying in degree of happiness [42]. Although children’s importance ratings were lower in general, they still provided insight into the relative importance of determinants according to children.” [Discussion, line 369 - 377]
Comment 2.8
Discussion - overall provided nice context. Streamlining it a bit would help the main ideas come thru better.
Response 2.8
We thank the reviewer for complimenting us on the nice context of our discussion. We double checked the streamlining of our discussion and main ideas. We additionally asked the language editor to pay attention to this while editing. All changes can be found in the revised version of the manuscript.
Comment 2.9
Child perceived barriers (ages 11-12) to sleep
https://www.ncbi.nlm.nih.gov/pmc/articles/PMC5373486/
Response 2.9
We thank the reviewer for suggesting this previous research work and incorporated it in the discussion of our manuscript.
This now reads…
“Shortcomings in children’s sleep environment, including too much light, not the right temperature, noise, uncomfortable sleeping place, uncomfortable sleep materials, and distractions in the bedroom, were identified as potential determinants. This finding is partly in line with Bagley et al. who found that temperature and noise were associated with more sleep-wake problems in 10-13 year olds, while the amount of light was not related to any of their sleep outcomes [25]. The review by Allen et al. also showed limited support for adjusting the bedroom light to as dark (i.e. only two studies) and as quiet (i.e. only two studies) as possible for better sleep [17]. In contrast, a recent review of the WHO found evidence for a relationship between ambient noise and inadequate sleep [40]. Also, a previous study (2017) where 11-12 year old multi-ethnic adolescents identified sleep-disturbing household activities, found that disorganization in the home environment such as TV or noise disturbance, family members phone calling, or night-time home visitors, were related to disturbed sleep [41]. In conclusion, environmental factors can be sleep-disrupting for some children, depending on children’s individual preferences for the amount of light, room temperature, noise level, and other distractions.” [Discussion, line 328 – 344]
- Spilsbury, J. C.; Patel, S. R.; Morris, N.; Ehayaei, A.; Intille, S. S., Household chaos and sleep-disturbing behavior of family members: results of a pilot study of African American early adolescents. Sleep health 2017, 3, (2), 84-89.
[References, line 748 - 750]
Comment 2.10
There are multiple measures of children’s self-reported sleep:
https://www.ncbi.nlm.nih.gov/pmc/articles/PMC5303893/
Response 2.10
We are aware that there are self-report measures for children to self-report on their sleep. We agree with the reviewer, based on his/her previous comments, that it would have been interesting to have administered some sleep screener to the sample. Unfortunately, we do not have this information and therefore this is added as limitation to the discussion section of our manuscript.
This reads…
“Lastly, we did not screen the participating children and parents on potential sleep problems or other medical conditions. There is a possibility that poor sleepers and parents whose children have trouble sleeping were more interested in participating in this study despite the emphasis on recruiting children and parents with both adequate and inadequate sleep health.” [Discussion, line 364 - 368]
Comment 2.11
Figure 1- this is a poor quality figure.
Response 2.11
We updated the resolution of Figure 1 and uploaded a high-resolution figure.

Reviewer 3 Report
no comments
Author Response
We thank reviewer for his/her time to review our revisions and their valuable remarks in this second review round. We revised the manuscript and integrated the feedback . Below we provide a short reply to the Reviewer’s comment.
We thank the reviewer for noting a minor spell check. Therefore, we have sent our manuscript to a language editing service for detailed corrections. All corrections are highlighted in the manuscript using track changes.
